# Mapping the gene network landscape of Alzheimer's disease through integrating genomics and transcriptomics

Sara Brin Rosenthal[1]*, Hao Wang[2]*, Da Shi[3,4], Cin Liu[2], Ruben Abagyan[3], Linda K. McEvoy[2,5], Chi-Hua Chen[2]*

1 Center for Computational Biology and Bioinformatics, University of California San Diego, La Jolla, California, United States of America, 2 Center for Multimodal Imaging and Genetics, Department of Radiology, University of California San Diego, La Jolla, California, United States of America, 3 Skaggs School of Pharmacy and Pharmaceutical Sciences, University of California San Diego, La Jolla, California, United States of America, 4 Biomedical Informatics and Data Science Directorate, Frederick National Laboratory for Cancer Research, Frederick, Maryland, United States of America, 5 Herbert Wertheim School of Public Health and Human Longevity Science, University of California San Diego, La Jolla, California, United States of America

☯ These authors contributed equally to this work.
* sbrosenthal@ucsd.edu (SBR); haw040@ucsd.edu (HW); chc101@ucsd.edu (C-HC)

**Data Availability Statement:** All relevant data are within the paper, its Supporting Information files, and on Zenodo at https://zenodo.org/record/

## Abstract

Integration of multi-omics data with molecular interaction networks enables elucidation of the pathophysiology of Alzheimer's disease (AD). Using the latest genome-wide association studies (GWAS) including proxy cases and the STRING interactome, we identified an AD network of 142 risk genes and 646 network-proximal genes, many of which were linked to synaptic functions annotated by mouse knockout data. The proximal genes were confirmed to be enriched in a replication GWAS of autopsy-documented cases. By integrating the AD gene network with transcriptomic data of AD and healthy temporal cortices, we identified 17 gene clusters of pathways, such as up-regulated complement activation and lipid metabolism, down-regulated cholinergic activity, and dysregulated RNA metabolism and proteostasis. The relationships among these pathways were further organized by a hierarchy of the AD network pinpointing major parent nodes in graph structure including endocytosis and immune reaction. Control analyses were performed using transcriptomics from cerebellum and a brain-specific interactome. Further integration with cell-specific RNA sequencing data demonstrated genes in our clusters of immunoregulation and complement activation were highly expressed in microglia.

## Author summary

Alzheimer's disease (AD) is recognized as the leading primary cause of dementia, resulting in a high socioeconomic burden. Understanding the disease pathogenesis serves as the cornerstone of exploring potential drug targets, therapeutic strategies and clinical intervention. As a complex disease, the development of AD involves pathological changes in multiple biological processes, and is impacted significantly by genetic factors. Through

5786722#.Ybtti73MKC8 (DOI: 10.5281/zenodo.
5786722).

**Funding:** CHC was supported by funding under
R01MH118281, R56AG061163 from National
Institutes of Health. The funders had no role in
study design, data collection and analysis, decision
to publish, or preparation of the manuscript.

**Competing interests:** The authors have declared
that no competing interests exist.

integration of the available genomic, protein-protein interactions (interactomic) and transcriptomic data, we identified a disease gene network that includes a total of 788 genes, and annotated 17 major gene clusters which encompassed the main categories of biological pathways with reported alterations in AD. The results revealed a landscape of AD etiology, with major pathological changes that extend from gene transcription and RNA metabolism, proteostasis, lipid metabolism, immune reactions to synaptic dysfunction. The systems-level approach of the present study can also be applied to other complex diseases with a significant genetic component.

## Introduction

Late Onset Alzheimer's disease (AD) is a neurodegenerative disorder recognized as the leading primary cause of dementia with a heritability estimate of 50–80% [1,2]. The advent of large-scale genome-wide association studies (GWAS) has revealed associations between single nucleotide polymorphisms (SNPs) and risk of AD, allowing for new insight into the genetic basis of this disease. The latest large AD GWA studies have identified over 40 risk loci [3–9].

Beyond GWAS, recent large transcriptomic studies have also begun to yield converging findings on differentially expressed genes associated with AD. The success of gene discovery combined with network-based approaches to human diseases [10], leads to opportunities for elaboration of AD pathophysiology at multiple omic levels via embedded molecular interactions (interactome). A key tenet of this approach is to address the overall dysfunction of disease genes within the context of their molecular interactions. Interactomes enable study of collective biological interactions together with basic units of protein-protein interaction, and mediators of various intracellular signaling and regulation [11]. This approach has been successfully used to provide novel insight into many diseases [10,12,13].

Emerging evidence from systems biology and network studies has revealed the association between AD and an increasing number of molecular networks, most notably including lipid metabolism with the *APOE* ε4 allele, and immunological dysfunction involving microglial cells [14–20]. These studies identified gene networks using transcriptomic or proteomic data but integration with AD GWAS is lacking. AD GWAS have made substantial progress identifying multiple SNPs associated with AD risk that have been replicated across studies [4,5]. Genetic variations captured by GWAS are inherited, and are generally not confounded by secondary changes from disease progression, whereas transcriptomics provides information on factors that are both inherited and non-inherited (e.g. affected by environmental exposure and comorbidities). These factors can reflect dynamic and tissue-specific patterns, such as the distinctive neural involvement at a late stage of life in AD [21]. One prior study did integrate AD GWAS with transcriptomics [22]. However, normal rather than AD brains were used in the analyses, and the sample size of AD GWAS was small (17,008 cases and 37,646 controls), compared to the two recent larger AD GWAS with AD-by-proxy samples in our main analysis (71,880 proxy cases and 383,378 controls) [4] and ancillary analysis of autopsy-documented AD samples (35,274 cases and 59,163 controls) [5].

We further leverage the power of the interactome to reveal AD pathophysiology by identifying the significantly proximal neighbors of AD GWAS genes in the interactome to form an expanded AD disease module. The rationale for including proximal genes is that previous studies showed enrichment of new disease loci or drug targets often among interactome neighbors of existing GWAS genes [19,23,24]. We then integrate data of transcriptomic dysregulation in AD brains with the expanded AD disease module, considering the transcriptomic data

from both the temporal cortex as major disease-relevant tissue, and the cerebellum as the hypothesized control, based on the evidence that tau pathology, the pathological feature of AD that closely associated with clinical symptoms, emerges in temporal cortex early in the course of the disease before spreading to the other cortical areas in later stages, with minimal involvement to the cerebellum [25–29]. Furthermore, application of multiscale community detection enables identification, annotation and mapping of the hierarchical substructure of biological pathways associated with the AD gene network [30].

## Results

### AD disease module

Our analysis workflow is illustrated in **Fig 1**. 192 AD risk genes were identified in a large AD-by-proxy GWAS, which identifies both AD patients and individuals with parental history of AD as cases [4]. In the AD-by-proxy GWAS, these risk genes were selected by using positional, expression quantitative trait loci (eQTL) and chromatin information, through the functional mapping and annotation (FUMA) tool [31], on AD GWAS summary statistics [4]. Out of the 192 genes, 142 were found in the interactome (STRING database) and included in our subsequent analyses. As genes not found in the interactome are poorly characterized, they were excluded. New genes with no known information of interaction were excluded as well (**Table D** in **S1 Data**).

The 142 AD associated risk genes were significantly localized in the interactome, compared with random gene sets ($p = 3.8 \times 10^{-9}$), with 59 of 142 risk genes (42%) directly connected with at least one other risk gene (i.e., interconnected risk genes). The largest connected component consisted of 14 risk genes (**Fig A** in **S1 Text**). We took node degree into account for the selection of random gene sets to control for potential bias that disease genes may have a high degree of connectivity, resulting from the fact that they are well-studied rather than from their biological properties. We also replicated our analysis using a brain-specific network from GIANT [24]. 191 out of 192 AD risk genes were present in this interactome and also significantly localized compared to random gene sets ($p < 10^{-16}$).

### Expanding the AD disease module

As a quantitative way of expanding the AD module to include genes proximal in network space, we used network propagation, a tool which has repeatedly driven novel biological discoveries (**Fig 1**) [32]. By seeding the network propagation algorithm with the 142 AD risk genes, we identified an expanded AD disease module of 788 genes significantly proximal to one or more seed genes (including 142 seed genes and 646 proximal genes, **Fig 2**). The genes in the AD disease module, including both seed genes and proximal genes were highly enriched for genes related to synaptic function, especially abnormal synaptic transmission, based on mouse knockout data (**Fig 3**) [33,34]. Using the 646 proximal genes alone, the gene-disease enrichment analysis also demonstrated that AD was the top enriched disease (**Fig F** in **S1 Text**).

A recent AD GWAS of diagnosed AD patients identified 400 AD-related genes, with varying levels of support, such as functional consequence, eQTL, tissue expression [5]. Of these 400 genes, 103 were also identified in the expanded AD module. Note that most of these were also found in the AD-by-proxy GWAS [4]; since the data partially overlap between the two GWAS, this is not unexpected. However, there were 26 proximal genes identified by our network, but were identified in the more recent AD GWAS [5]. This overlap is highly significant (OR = 2.0, $p = 0.0008$, Fisher's exact test). We note that although some of the underlying data overlap, our network analysis approach successfully identified genes missed by the first GWAS [4].

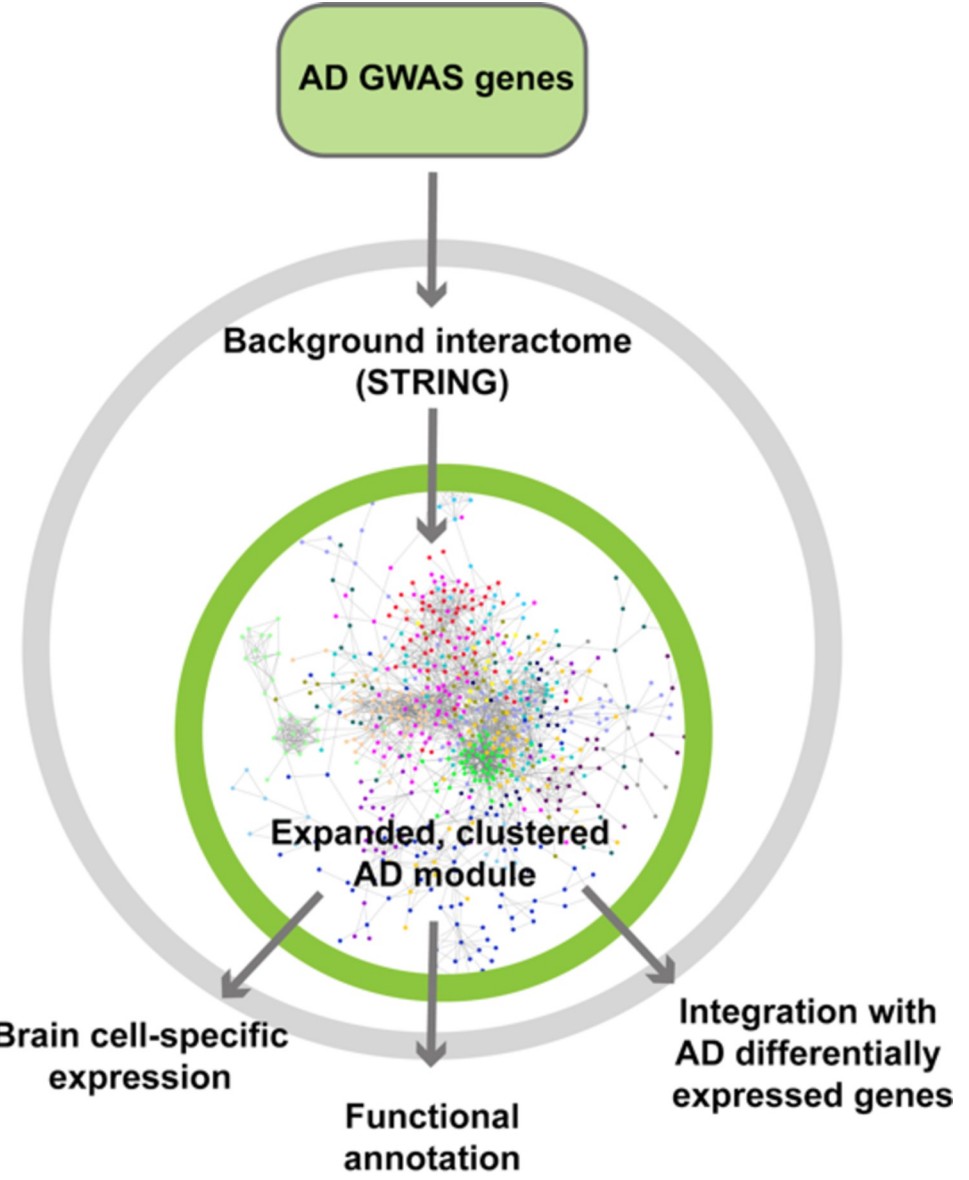

**Fig 1. Analysis workflow.** 142 GWAS-identified AD genes were included as the seed genes of the AD disease module, and propagated to an expanded network consisting of another 646 predicted proximal genes based on the background protein-protein interactome (STRING). The expanded AD module was used for further functional annotation, pathway clustering and gene expression analyses integrating with transcriptomic data.

We sought to interpret the structure and content of the expanded AD module, to identify major biological pathways and functions represented in the disease. For this purpose, we applied a graph-based clustering algorithm to the expanded AD module [35], which revealed a strong clustering structure in the network, identifying 18 distinct clusters of 10 genes or more (**Table 1**). Additionally, we found 15 smaller clusters containing 2–9 genes, and 15 orphan genes forming their own clusters. We focused our analysis on the 18 largest clusters with at least 10 genes (**Fig 2**). An alternative clustering strategy, multiscale community detection, revealed largely similar clusters (**Fig 4** and **Table C** in **S1 Data**).

Functional annotation of these clusters demonstrated significant association with multiple well-characterized biological pathways. In addition to the many clusters that represent

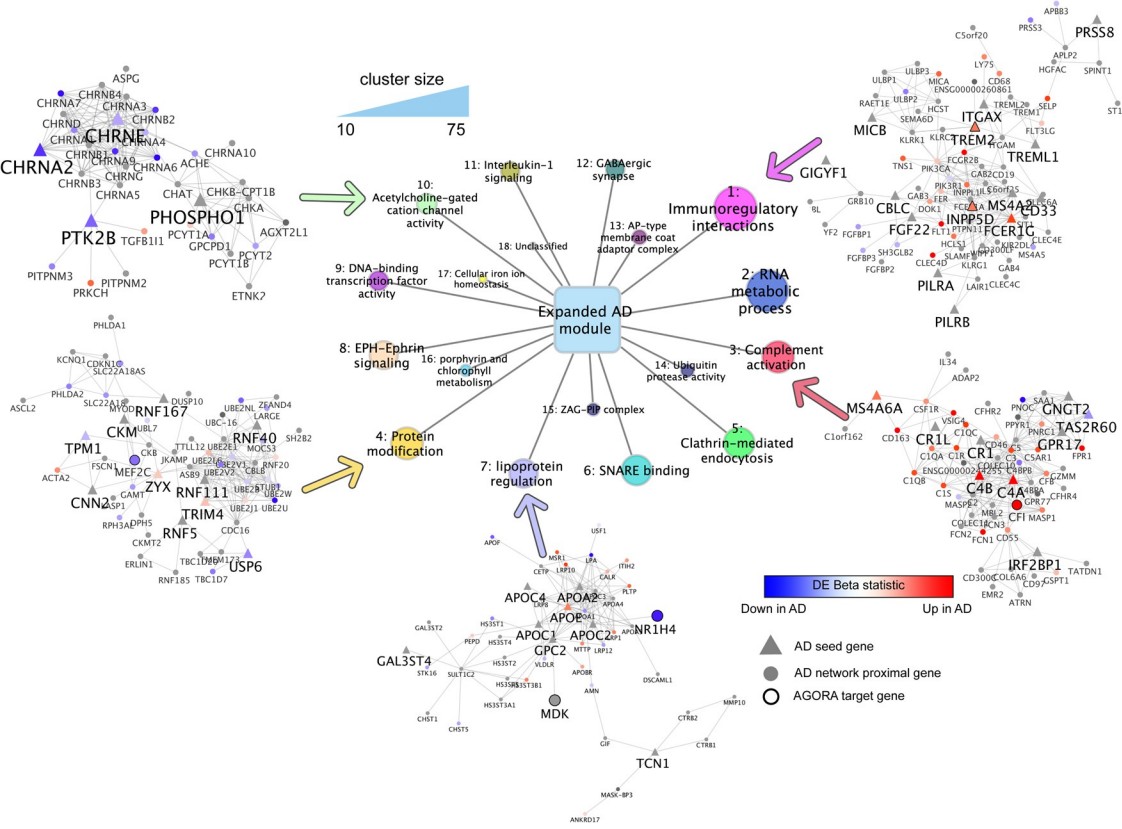

**Fig 2. Expanded AD disease module and highlighted distinctive clusters annotating different biological pathways possibly involved in AD.** Clusters identified in the expanded AD module labeled in the center. Genes comprising 5 selected clusters are depicted around the edges of the figure. The temporal cortex RNAseq beta statistic is mapped to the gene color, and shape indicates AD GWAS seed gene (triangle) or network proximal gene (z > 2, circle). Bold black outlines indicate AGORA proposed drug targets (e.g. CHRNA2, PTK2B). Seed genes or genes with AGORA targets are labeled with a larger font, while other genes identified by network propagation are labeled with a smaller font.

previously reported mechanisms in AD pathogenesis (**Table 1** and **B** in **S1 Data**) [36–48]. we also identified pathways whose roles in AD are as yet unclear, including zinc α2-glycoprotein-prolactin-inducible protein (ZAP-PIP) complex as well as porphyrin and chlorophyll metabolism. Clusters highlighted in **Table 1** and **Fig 2** with significant clinical implications include immunoregulation (cluster 1, $p = 1 \times 10^{-12}$), complement activation (cluster 3, $p = 5 \times 10^{-33}$), RNA metabolism (cluster 2, $p = 7 \times 10^{-17}$), acetylcholine-gated cation channel activity (cluster 10, $p = 6 \times 10^{-22}$) and GABAergic synapse (cluster 12, $p = 1 \times 10^{-5}$). Based on correlation in biological functions, the clusters were further grouped into 5 classes that cover various aspects of AD pathophysiology, including immune reactions, gene transcription and RNA metabolism, proteostasis, synaptic function and substance metabolism (**Table 1**). We note that observing highly significant associations with known biological pathways is expected, as genes within the same pathway are likely to interact. Enrichment *p*-values indicate confidence of the observed associations. Identified clusters contain both AD GWAS genes, as well as proximal genes in the expanded AD module (**Tables A** and **B** in **S1 Data**).

## Transcriptomic dysregulation proximal to AD genes

To further characterize AD-proximal genes, we used data from the Mayo Clinic RNAseq study. These data identify differential expression of genes in AD patients compared to controls

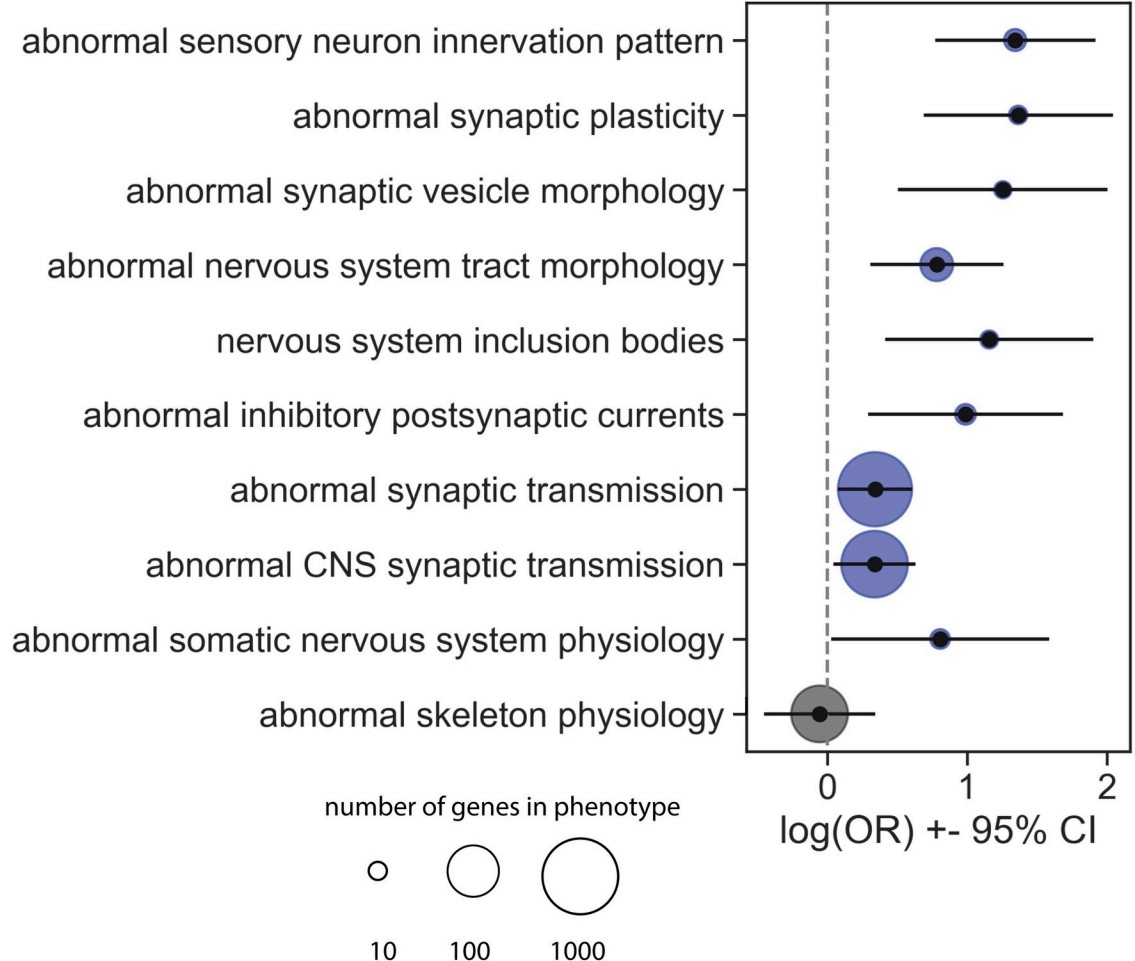

**Fig 3. Annotations of Mammalian Phenotype Ontology on the AD network genes.** The figure shows the odds ratios and 95% confidence intervals for the 9 significantly enriched brain-related phenotypes, and one negative control phenotype (abnormal skeleton physiology). The circle size indicates the number of genes which result in that phenotype when knocked out in mice.

from the temporal cortex (**Fig B** in **S1 Text**) and cerebellum [49]. Of 1,213 genes that were significantly upregulated in the temporal cortex (adjusted $p < 0.05$, beta $> 0.5$), 85 were also found in the expanded AD module (hypergeometric $p = 0.002$). Downregulated genes were not similarly enriched (hypergeometric $p = 0.5$). This effect was even more pronounced when we examined the full distribution of network proximity z-scores of the up-regulated genes, with this gene set having significantly higher z-scores than 13,135 genes not up-regulated in AD ($p = 2\times10^{-14}$, K-S 2-sample test), as shown on **Fig C** in **S1 Text**. This was in contrast to the data from cerebellum, where there was no significant overlap of up-regulated genes (hypergeometric $p = 0.116$), or difference in z-scores between up-regulated genes and the rest in the network ($p = 0.069$) (**Fig D** in **S1 Text**).

Clusters with consistent up- or down-regulation of genes were identified by integration of the RNA sequencing (RNAseq) data of genetic differential expression in the temporal cortex. Eight clusters demonstrated some evidence of dysregulation (Benjamini-Hochberg FDR $< 0.2$, hypergeometric test; **Table 1** and **B** in **S1 Data**). In particular, cluster 10 annotated for acetylcholine-gated cation-selective channel activity was down-regulated in AD compared to healthy controls, with 7 significantly downregulated genes, relative to only 1 upregulated

**Table 1. Clusters in the expanded AD disease module.**

| Cluster* | Number of genes | Enrichment of dysregulated genes in AD cortex (BH FDR) | Pathway p-value | Function |
|---|---|---|---|---|
| Class 1. Immune reactions | | | | |
| 1 | 75 | ns† | $1.13\times10^{-12}$ | Immunoregulatory interactions between a lymphoid and non-lymphoid cell |
| 3 | 55 | $6.80\times10^{-5}$ | $4.84\times10^{-33}$ | Complement activation |
| 8 | 48 | ns | $3.42\times10^{-26}$ | EPH-Ephrin signaling |
| 11 | 31 | 0.08 | $2.48\times10^{-8}$ | Interleukin-1 signaling |
| 15 | 20 | 0.06 | 0.00017 | ZAG-PIP complex |
| Class 2. Gene transcription and RNA metabolism | | | | |
| 2 | 73 | ns | $6.76\times10^{-17}$ | RNA metabolic process |
| 9 | 33 | 0.06 | $1.36\times10^{-13}$ | DNA-binding transcription factor activity |
| Class 3. Vesicular transport, post-translational protein modifications, trafficking and proteostasis | | | | |
| 4 | 55 | ns | $6.30\times10^{-14}$ | Protein modification by small protein conjugation |
| 5 | 53 | ns | $1.56\times10^{-46}$ | Clathrin-mediated endocytosis |
| 6 | 52 | ns | $2.88\times10^{-15}$ | SNARE binding |
| 13 | 24 | ns | $1.06\times10^{-15}$ | AP-type membrane coat adaptor complex |
| 14 | 21 | ns | $2.49\times10^{-7}$ | Ubiquitin-like protein-specific protease activity |
| Class 4. Synaptic function | | | | |
| 10 | 33 | 0.18 | $5.81\times10^{-22}$ | Acetylcholine-gated cation-selected channel activity |
| 12 | 31 | ns | $1.18\times10^{-5}$ | GABAergic synapse |
| Class 5. Substance metabolism | | | | |
| 7 | 49 | 0.18 | $6.05\times10^{-16}$ | Regulation of plasma lipoprotein particle levels |
| 16 | 18 | ns | $7.87\times^{-12}$ | Porphyrin and chlorophyll metabolism |
| 17 | 11 | 0.18 | $3.03\times10^{-12}$ | Cellular iron homeostasis |

*Cluster 18 genes were unclassified by functional annotation and not included.

†ns: not significant.

gene (FDR = 0.18, hypergeometric test). These include *CHRNA2*, and *PTK2B*, both of which were implicated at the GWAS level, as well as the transcriptomic level. Cluster 3, annotating for complement activation, was strongly upregulated in AD, with 20 significantly up-regulated genes, compared to only 2 downregulated genes ($p = 6.8\times10^{-5}$, hypergeometric test). The upregulated genes include *C4B* and *C4A*, which were AD-GWAS genes, as well as *CFI*, which was not implicated by the GWAS, but is found in a list of expert-curated potential AD targets (Agora database). Importantly, we found that randomly selected gene clusters with similar properties to the AD gene clusters are much less dysregulated in the RNAseq data, with only one random gene cluster marginally enriched for AD differentially expressed genes (FDR < 0.2). This suggests that the network proximal genes to AD GWAS hits are more strongly dysregulated than randomly selected regions of the interactome, pointing to a possible link between genomic variants and transcriptomic dysregulation in network proximal biological pathways.

## Cell-specific preferential expression

Cross-referencing with brain tissue expression data demonstrated unique patterns of cell-specific gene expression across the identified clusters, with clear association with their annotated biological functions (**Fig 5A** and **E** in **S1 Text**). In particular, genes in clusters 1 (immunoregulation) and 3 (complement activation) were preferentially expressed in microglial cells ($p = 1\times10^{-4}$ and $9\times10^{-3}$ respectively, Wilcoxon rank-sum test). A trend (not statistically

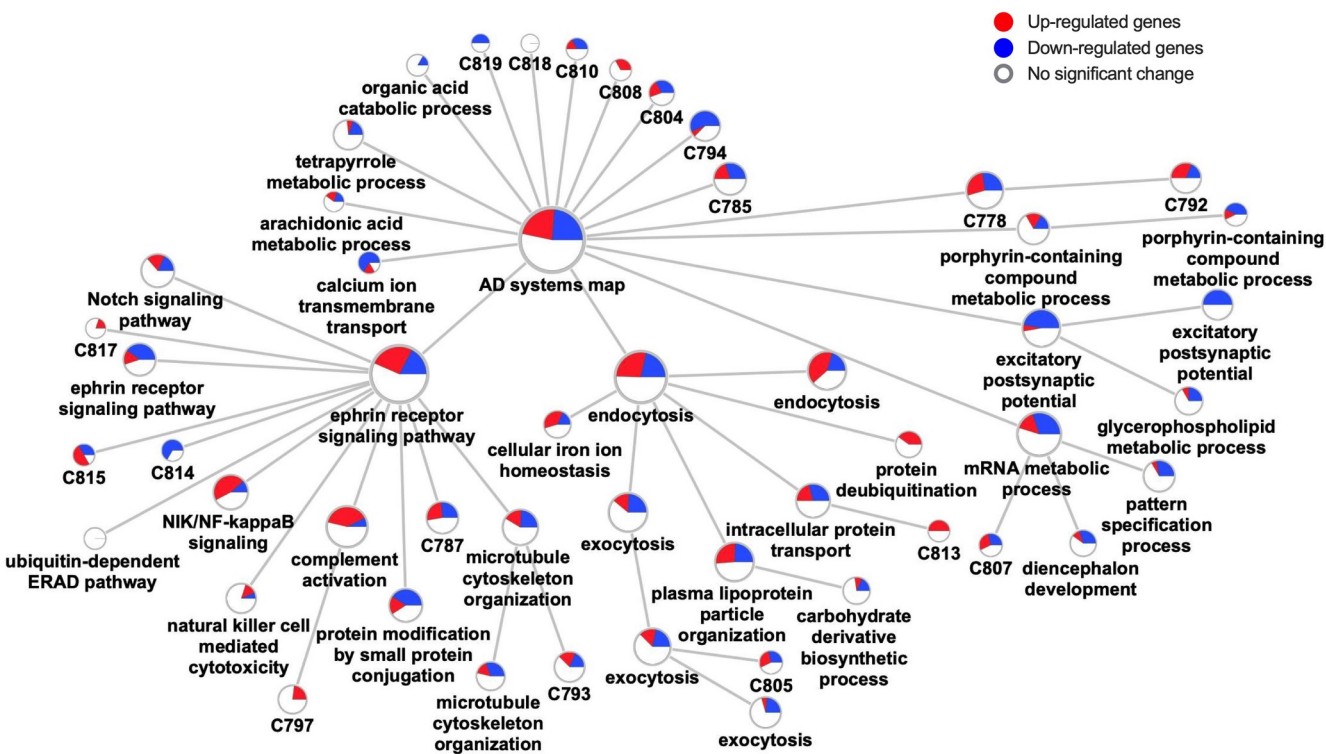

**Fig 4. Hierarchical graph of the AD gene network.** Pie charts indicate fractions of up- (red) and down- (blue) regulated genes.

significant) was also observed that genes in cluster 5 (Clathrin-mediated endocytosis) were preferentially expressed in oligodendrocytes ($p = 0.73$), in cluster 10 (acetylcholine-gated cation-selective channel activity) in mature astrocytes ($p = 0.64$), and in cluster 12 (IL-1 signaling) mostly in neurons ($p = 0.25$) followed by mature astrocytes ($p = 0.25$).

## Discussion

There was significant clustering among the 788 identified network genes, forming a distinct disease module in the interactome. The expanded AD module was enriched for genes involved in several biological pathways implicated in AD, including lipid metabolism, the immune system, endocytosis, the cholinergic and GABAergic pathways of the central nervous system (CNS), which are also annotated in the AD-by-proxy GWAS paper as expected. Our hierarchical network analysis revealed the multiscale structure of the AD gene network.

By integrating the AD interactome module with information on differential RNA expression between AD patients and healthy controls, we identified clusters of genes within the AD module that were primarily up- or down-regulated at the transcriptomic level [49]. Significant enrichment of dysregulated genes in AD was observed in 8 out of the 17 annotated clusters. Possible reasons could lead to the discrepancies between the transcriptomic and GWAS data in the other 9 clusters, such as that genes with small effects were not consistently identified in the two data sets, and/or there are more complicated epigenomic mechanisms involving differential expression of genes that were not detected by GWAS.

Finally, we further profiled cell-specific gene expression of these clusters with data from RNA sequencing of purified cells [49]. We select potentially clinically-relevant clusters to discuss in detail as follows.

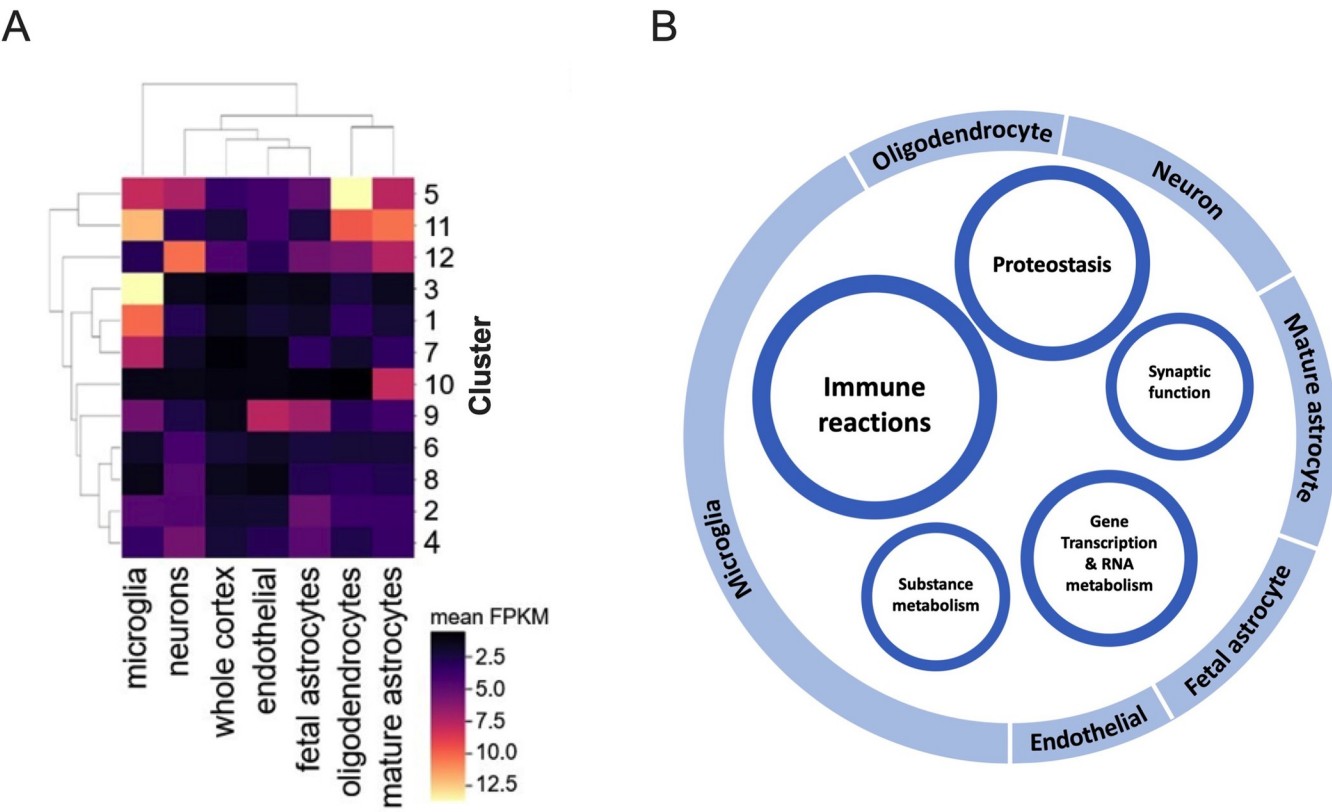

**Fig 5. Cell-specific gene expression across clusters and schematic model.** A) Mean expression (FPKM) from brain cell-types averaged across genes in each cluster (functional annotations for identified clusters are 1: Immunoregulatory interactions between a lymphoid and non-lymphoid cell, 2: RNA metabolic process, 3: Complement activation, 4: Protein modification by small protein conjugation, 5: Clathrin-mediated endocytosis, 6: SNARE binding, 7: Regulation of plasma lipoprotein particle levels, 8: EPH-Ephrin signaling, 9: DNA-binding transcription factor activity, 10: Acetylcholine-gated cation-selected channel activity, 11: Interleukin-1 signaling, 12: GABAergic synapse). B) Classes of identified pathways that are functionally related are presented as inner circles, with circle size roughly indicating relative class sizes. The overlying outer circle illustrates the types of cells in the central nervous system with overall preferential gene expression in each major class (e.g., microglia overlay the classes of Immune reactions and Substance metabolism because genes in multiple clusters of these two classes were highly expressed in microglia).

## The Cholinergic pathway

Involvement of the cholinergic system in AD has been known since the 1970s [43]. Our data-driven approach identified a cluster of 33 genes (**Table A** in **S1 Data**), the majority of which are implicated in cholinergic function [43]. Consistent with decreased cholinergic activity in AD, genes in this cluster were strongly down-regulated in AD compared to the control group, with 7 significantly down-regulated genes, and only 1 up-regulated gene. This cluster includes 15 highly interconnected genes encoding receptor subunits (*CHRNA*, *CHRNB*, *CHRND*, *CHRNE* and *CHRNG*) for the excitatory neurotransmitter acetylcholine, which is strongly involved in memory function [43]. This cluster also contains genes involved with the synthesis (*CHAT*, *CPT1B*) or breakdown (*ACHE*) of acetylcholine. Interestingly, *PTK2B* is a member of this cluster, which was identified from the GWAS [4]. It encodes PTK2B, a tyrosine kinase that is involved in regulation of long-term potentiation in the hippocampus as a likely neural substrate of memory formation [50]. *PTK2B* was also significantly down-regulated in AD compared to controls. While no longer believed to be the primary pathologic mechanism underlying AD, these results show that cholinergic dysregulation and disruption of hippocampal signaling are important factors in AD pathogenesis.

## Complement activation

Emerging evidence suggests the importance of neuroinflammation in the pathogenesis of AD. Complement activation has been observed in the brain tissue of AD patients and seems to contribute to an important local inflammatory state, with increased expression of *C4* observed in AD patients. Other members (*C3* and *C1q*) of the pathway are also implicated in AD, implying a role for the entire complement activation cascade in AD pathogenesis [51]. Coding genes for these proteins, such as *C4*, *CR1*, *CR1L*, were identified from GWAS and the majority showed up-regulated expression patterns in our cluster.

From the cell-specific data, we found genes in this cluster highly expressed in microglial cells (**Fig 5A**). A mechanistic cascade has been suggested involving increased expression of *C3* in microglia and astrocytes by amyloid β (Aβ) oligomers, which tag synapses, promotes recruitment of microglia, and mediates elimination of the tagged synapses [52]. Given that these clustered genes are associated with AD risk in GWAS, up-regulated in AD, and preferentially expressed in cells with close etiological relationship with AD, these findings suggest that the genomic and transcriptional alterations in this cluster likely play a causal role in AD pathogenesis, rather than being secondary consequences of disease progression, as previously suggested [53].

## Immunoregulatory interactions and ZAG-PIP complex

The crucial role that microglia-related immunoregulation plays in AD is supported by our network analyses and the preferential expression of these genes in microglia (**Fig 5A**) [54,55]. Some of the key genes in our study are consistent with current knowledge in this field. *TREM2* is an up-regulated cell surface receptor in AD, and exclusively expressed in immune cells including macrophages, dendritic cells and microglial in CNS [54]. The coded protein of this gene has been found to be involved in activation of microglia via its soluble fragment around Aβ plaques, maintaining plaque morphology and neurotoxicity [56,57]. In addition, important genes in the phosphatidylinosol-3 kinase (PIK3) pathway that are closely related to downstream effects of *TREM2*, such as *INPP5D*, *INPPL1* and *PIK3CA*, were also universally up-regulated in our study.

We observed a cluster of genes associated with the ZAG-PIP complex (**Table A** in **S1 Data**), whose role in AD is not yet fully described. Although both proteins are multifunctional, being involved with such processes as immunoregulation, fertilization and lipolysis [58,59]. ZAG is structurally similar to a truncated secretory MHC-I-like protein, and interacts with PIP which resembles a light chain and binds to both CD4 and IgG [60]. The *ZAG* (also known as *AZGP1*) gene has been identified as significant for AD by both reference GWASs used in the present study [4,5]. We observed up-regulation of *ZAG* but overall down-regulation of other genes in this cluster among AD patients, implying possible dysregulation of immune reactions associated with this complex in the disease.

## Lipoprotein regulation

The Apolipoprotein E (*APOE*) gene, the strongest genetic risk factor for late-onset AD [61], was upregulated in our study. APOE is a lipid transport protein that transports cholesterol and lipids to neurons through the low-density lipoprotein (LDL) receptor family for use in cell membrane maintenance and neuronal repair [41].

In our disease module, *APOE* was located within a cluster of 49 genes (**Table A** in **S1 Data**) that are mainly involved in lipid metabolism, including *APOA*, *APOB*, *APOC*, and *APOF* genes that showed mixed patterns of up- or down-regulation. The *APOA* genes, which code for proteins in the high-density lipoprotein, were all down-regulated in AD whereas the

*APOBR* and most of the *APOC* genes were up-regulated. Most of the LDL receptor encoding genes in this cluster were up-regulated, but the very low-density lipoprotein receptor gene was down-regulated. Also included in this cluster are genes that code for heparan sulfate biosynthetic enzymes (HS3ST) which are thought to facilitate binding of APOE to the LDL receptor–related protein 1 (LRP1) [62]. Most of these genes (4 out of 6) were down-regulated. Other genes in this cluster are related to the immune system (*ANKRD17*, *NR1H4*, *TCN1*) and to inflammation (*CHST1*, *CHST5*, *GPC2*, *MDK*, *MMP10*, *MSR1*).

## RNA metabolism

AD has been associated with reduced levels of RNA-binding proteins (RBP), which are crucial in RNA metabolism and maintaining liquid-liquid phase separation for dynamic formation of supramolecular assemblies [63]. Disrupted homeostasis of RBPs leads to increased propensity of aggregation and sequestration of RBPs by abnormal RNAs, and subsequent formation of neurotoxic deposits such as Tau proteins [64]. In addition to genes identified by previous GWAS (*SCARA3*, *GEMIN7*, *TSC22D4*, *TAF6*, *CPSF2*), additional RBP genes, involved in multiple stages of the RNA life cycle, were found in our AD disease module (*ZC3HAV1*, *DNAJC2*, *SYMPK*, *C8orf34*, *CPSF3*, *HEXIM2*, *HOXA5*, *ESX1*, *HOXA10*, *SOX3*, *HOXB5*, *LARP7*), majority of which were significantly down-regulated in AD (**Table A** in **S1 Data**).

## Proteostasis

Proteostasis, or protein homeostasis, involves clearing aberrant, mis-localized or excessive proteins, the imbalance of which is evident in many neurodegenerative disorders [65]. We identified a large class of functionally correlated biological pathways that are associated with this process, including post-translational modifications, intracellular trafficking, vesicular transport and degradation of protein (**Table 1**).

Consistent with published evidence on impaired ubiquitination-proteasome system (UPS) in AD due to ineffective protein clearing [66], a large set of relevant genes was enriched in our disease module, the majority of which were down-regulated in AD. Interestingly, associated genes of UBE2V1-UBE2N and UBE2V2-UBE2N heterodimers (*UBE2V1*, *UBE2V2* and *UBE2NL*) were also found among this cluster, which catalyzes synthesis of non-canonical polyubiquitination that does not lead to degradation by proteasome but rather downstream inflammatory responses [67], suggesting the mechanisms of UPS dysfunction in the development of AD are likely more complicated. In addition, we identified multiple significantly down-regulated deubiquitinating enzyme genes in AD, suggesting a role for the imbalance of ubiquitination in AD pathophysiology.

Interestingly, the clustered genes involved in Clathrin-mediated endocytosis were expressed preferentially in oligodendrocytes (**Fig 5A** and **E** in **S1 Text**). Although most studies on the association between dysregulated endocytosis and the formation of Aβ proteins focused on the neurons [39], the above observation may imply that oligodendrocytes are also involved. Furthermore, as differentiation and signaling of oligodendrocytes rely on endocytosis for internalization of transferrin, an iron transporter [68], it is possible that the well-known iron dyshomeostasis in AD affects not only neurons, but also oligodendrocytes and oligodendrocyte progenitor cells [48].

## Summary of molecular pathways in AD

Based on the results from our AD gene network and pathway annotation, we propose 5 classes of functionally related molecular pathways that are significantly associated with AD: 1) immune reactions, 2) gene transcription and RNA metabolism, 3) vesicular transport, post-

translational protein modifications, trafficking and proteostasis, 4) synaptic function (including acetylcholine-gated channel activity and GABAergic synapse) and 5) substance metabolism (including lipid, iron, porphyrin and chlorophyll) (**Table 1**). We noted that the classification above can be changed slightly because some pathways have multiple functions and can be involved in various classes. An example is the EPH-Ephrin signaling pathway, which is associated both with synaptic dysfunction and CNS immune dysregulation [42,69].

A schematic model of gene network in AD pathophysiology based on results of the present study is illustrated in **Fig 5B**. Although alterations in gene expression are complicated in many of the above classes, there is likely predominant up-regulation of immune reactions (IL-1 signaling and complement activation) and lipid metabolism, as well as down-regulation of cholinergic neurotransmission.

Our hierarchical network analysis generally verified the five classes proposed above, where the identified parent nodes included: 1) immune responses (though ephrin receptor signaling pathway was labelled and likely driven by the number of overlapping genes), 2) RNA metabolism, 3) endocytosis, 4) excitatory postsynaptic potential. The pathways for the class of substance metabolism proposed above were scattered in different nodes (**Fig 4** and **Table C** in **S1 Data**).

The above results demonstrate a landscape of AD etiology, and major pathological changes that range from molecular (gene transcription and RNA metabolism, proteostasis, and substance metabolism), cellular (immune reactions) to tissue-level (synapses) dysfunction. This approach can be applied to other complex diseases with significant genetic component. Future studies on modification and verification of the disease network may provide further insight on prevention and therapeutic intervention of AD.

## Potential limitations

The main limitation of the present study is that the dataset for AD risk genes from GWAS is incomplete, although it still covers a significant amount of information. Using network propagation can mitigate this issue as this method is an amplifier of genetic associations [32]. There are multiple choices of interactome, each with their own pros and cons, and many tissue-specific networks that can be obtained (e.g., GIANT [24]). We chose the STRING interactome based on the results of a systemic evaluation of the performance of diverse networks [70], although other interactomes may yield different results and should be considered in future studies. We also used a brain-specific network from GIANT to test consistency of the localization analysis. Our chosen networks from high throughput experiments or computational prediction are potentially less susceptible to the literature bias to well-studied genes. In addition, the present study did not include 50 GWAS-identified genes that are not in the STRING interactome (**Table D** in **S1 Data**). Their roles in AD etiology may also be important, and warrant further investigations.

## Materials and methods

### AD risk genes

To identify genes related to AD, summary statistics from a large-scale GWAS were used [4]. AD risk genes were derived from a GWAS of 455,258 individuals with 71,880 proxy cases and 383,378 controls [4]. This is one of the largest GWAS of AD to date, having identified 29 independent loci and 192 genes associated with AD. A second AD GWAS was used as a replication GWAS, and included with 35,274 clinical and autopsy-documented AD cases and 59,163 controls. This GWAS identified 400 candidate genes associated with AD [5].

## Molecular interaction network (interactome)

The STRING database of protein-protein interactions was selected as the background interactome for the analysis. STRING consists of both physical and functional interactions, derived through co-expression, biological knowledge databases, and computational techniques. Interactions are scored based on accumulation of different types of evidence [71]. In our analysis we used interactions classified as 'high confidence' (combined score > 0.7), for the human interaction version 10.5, containing 15,131 proteins and 359,776 interactions.

## Network propagation

A network propagation algorithm was used to explore the network proximity to a set of genes identified as significantly associated with AD [32]. Network propagation amplifies biological signals in networks, enabling exploration of genes significantly nearby in network space, and improving on simpler measures such as first nearest neighbors [32]. The network propagation algorithm simulates how heat would spread, starting from a set of 'hot' seed genes (GWAS-discovered seed nodes). In the simulation, the heat spreads from gene to gene along the interactions in the network. The result is a set of 'hot' genes, which are likely related to the starting seed genes in biological process or pathway, as the adjacent genes in a network likely have similar biological functions. This process enables identification of genes related to multiple seed genes rather than a single one, because more heat will accumulate in genes that are close to multiple seed genes. This process is described in the following equation [72]:

$$F^t = \alpha W' F^{t-1} + (1 - \alpha)Y$$

Where $F^t$ is the heat vector at time t, Y is the initial value of the heat vector, where the value of seed genes is 1/S, the value of non-seed genes is 0, and S is the total number of seed genes. W' is the normalized adjacency matrix, and $\alpha$ in (0,1) represents the fraction of total heat which is dissipated at every timestep. We chose an $\alpha$ value of 0.5, based on previous work which demonstrated that the propagation algorithm is not sensitive to the choice of $\alpha$ as long as $\alpha \geq 0.5$ [71]. We refer the reader to our jupyter notebooks (https://zenodo.org/record/5786722#.Ybtti73MKC8 (DOI: 10.5281/zenodo.5786722)) and the original publication for more details on network propagation [72].

We compared the network propagation z-scores to a null model to find genes which were significantly more proximal to the seed genes than would be expected by chance, defined as 2 standard deviations from the mean (z > 2), where a z score of 2 corresponds to $p = 0.02$. We constructed a null model by selecting random sets of genes with similar degree distributions to the seed set, using the binning approach [73]. Nodes were grouped into bins where each bin had at least 10 nodes of similar degree. 5000 such gene sets were randomly selected to build up the null distribution. We computed a node-level z-score comparing the network proximity values from the seed set to the mean and standard deviation from the null model network proximities.

$$z_n = \frac{\log(F_{n,HC}) - \langle \log(F_{n,rand}) \rangle}{\sigma(\log F_{n,rand})}$$

Where $F_{n,HC}$ is the propagation score of gene n, for high confidence seed genes, $F_{n,rand}$ is the propagation score of gene n for randomly selected degree-matched genes, $<>$ denotes an average of gene n's propagation score over N randomly sampled sets, and $\sigma$ denotes the standard deviation of the random distribution. The proximity vectors were log transformed so they are approximately normally distributed.

## Network localization

We measured the localization of the AD gene set by calculating the number of edges shared between the genes in the focal set. This is similar to the 'significance' measure used in string-DB [71]. To measure significance, we calculated the localization of the full gene set, and compared this to the distribution of localization on 5,000 randomly selected, degree-matched gene sets of size equal to the number of disease risk genes. To ensure the localization is not dependent on a small number of hub nodes, we built up a distribution using a sampling procedure. We measured the number of edges connecting a randomly sampled set of 80% of the full set of AD genes, and compared to degree-matched random node set. We conducted 5,000 random samplings to build up the distribution, and did not find any effect from hubby genes (**Fig A panel B** in **S1 Text**).

## Gene enrichment analysis

Integration of the AD network with mouse knockout data was performed using the mammalian phenotype ontology, with data from the Jackson laboratory [74]. We identified 9 brain-related phenotypes which were significantly enriched for genes in the AD network, using a Fisher's exact test. A control phenotype (abnormal skeleton physiology) was not similarly enriched. We also evaluated the association of predicted 646 proximal genes with AD by gene-disease enrichment analysis (DisGeNET), using disgenet2r package for R [75].

## Clustering

Clusters of highly connected genes were identified in the AD network using a graph-based modularity maximization algorithm [35], commonly referred to as the Louvain algorithm, which iteratively identifies groups of genes which have many connections within the group and few connections between groups.

Functional enrichment of the gene clusters was conducted using the G:Profiler tool [76], using all genes in the full AD network as the background gene set, with adjustments for multiple tests (the Benjamin-Hochberg procedure). GO terms and KEGG and REACTOME pathways were tested for functional enrichment.

To verify the clustering results, we built an alternative hierarchical AD network by multi-scale community detection performed in Cytoscape using the CDAPS (Community Detection APplication and Service) application (29), with the HiDef community detection algorithm [77]. We used the 788 genes of the AD expanded disease module. Communities were annotated with significantly enriched GO terms and pathways from the G:Profiler tool.

## Agora database

Putative Alzheimer's disease genes that may be candidates for drug targets were downloaded from the Agora database on 1/9/2019 (https://agora.ampadportal.org/genes/). The database was contributed in part by the Accelerating Medicine Partnership–Alzheimer's Disease (AMP-AD) consortium.

## Transcriptomic study

To investigate whether a gene or pathway of the AD module is up or down regulated, we overlaid the differentially expressed genes provided by the Mayo Clinic RNAseq Study [49] from temporal cortex tissue and cerebellum of AD case-control post-mortem brains (84 cases and 80 controls) on the AD module to infer the effect direction of individual pathways. The data was accessed through the AMP-AD Knowledge Portal. The full AD network was tested for

enrichment of significantly up or downregulated genes (hypergeometric test), using a total of 14,631 genes that were found both in the interactome network and in the RNAseq data. The significance of the overlap between dysregulated RNAseq genes and genes in each cluster was also assessed (hypergeometric test). We compared the observed number of differentially expressed genes found in each cluster to the expected number given the observed 2,399 significantly differentially expressed genes (adjusted $p < 0.05$, abs(Beta) > 0.5). We applied the Benjamini-Hochberg procedure for multiple tests across all clusters. Additionally, we computed the enrichment of RNAseq dysregulated genes in random clusters, generated by seeding the network propagation algorithm with randomly selected seeds with a similar degree distribution as the AD seed genes, and then applying the graph-based clustering algorithm to the genes significantly proximal to these random genes. Thus, we found gene clusters with similar properties to the AD gene clusters, but which are not related to AD in any way. We computed the enrichment of the RNAseq dysregulated genes in these random gene clusters in the same way as for the AD gene clusters, and found only one gene cluster marginally dysregulated (FDR < 0.2), compared to 8 strongly or marginally dysregulated gene clusters in the expanded AD module (**Table 1** and **B** in **S1 Data**).

### Brain tissue specific analysis

Brain tissue expression data for neuronal, glial and endothelial tissue types were downloaded from the Brain RNA-seq Database [77]. FPKM values averaged over all genes per cluster were used to make the heatmap.

## Supporting information

**S1 Text. Supplementary Figures: Fig A: Network localization of AD GWAS genes.** (A) AD gene network identified by GWAS. Gene colors represent the differential expression beta statistic in the temporal cortex between AD and healthy controls. Edges represent high confidence interactions in the STRING database. (B) Distribution of number of edges interconnecting AD GWAS genes (blue) or randomly selected gene sets (yellow). 80% of the AD GWAS genes were sampled 5000 times to create the distribution. **Fig B: Heatmap of relative gene expression between AD patients and healthy control, temporal cortex.** This figure shows the top 100 most differentially expressed genes. Note that the patients (columns) were not clustered here- they are sorted by healthy and AD status. Only the genes (rows) are clustered. **Fig C: Transcriptomic study of AD genes in the temporal cortex.** (A) Overlap of up-regulated genes in the Mayo Clinic RNAseq data and the expanded AD disease module. (B) Significant difference of Z-scores between up-regulated and the rest of genes. **Fig D: Transcriptomic study of AD genes in the cerebellum.** (A) No significant overlap of up-regulated genes in the Mayo Clinic RNAseq data and the expanded AD disease module. (B) No significant difference of Z-scores between up-regulated and the rest of genes. **Fig E: Brain cell-specific mean expression (FPKM) of genes in identified clusters.** Functional annotations for the clusters are 1: Immunoregulatory interactions between a lymphoid and non-lymphoid cell, 2: RNA metabolic process, 3: Complement activation, 4: Protein modification by small protein conjugation, 5: Clathrin-mediated endocytosis, 6: SNARE binding, 7: Regulation of plasma lipoprotein particle levels, 8: EPH-Ephrin signaling, 9: DNA-binding transcription factor activity, 10: Acetylcholine-gated cation-selected channel activity, 11: Interleukin-1 signaling, 12: GABAergic synapse. **Fig F: Gene-enrichment analysis using predicted 646 proximity genes.** The plot shows ratio of proximity genes overlapped with each disease-related gene set in the available databases.
(DOCX)

**S1 Data.** Supplementary Tables: Table A: Annotation of genes in the AD disease module. Table B: Functional annotation of the genes in the AD disease module. Table C: Genes in the hierarchical network. Table D: GWAS-identified significant genes that are not in the STRING interactome.
(XLSX)

## Acknowledgments

The results published here are in whole or in part based on data obtained from Agora and the AD Knowledge Portal. Complete statements are detailed in their websites (https://agora. ampadportal.org/about/, https://adknowledgeportal.synapse.org/DataAccess/ AcknowledgmentStatements/). Mayo Clinic RNAseq Study with samples provided by Mayo Clinic Brain Bank and Banner Sun Health Research Institute (https://www.synapse.org/#!Synapse:syn20818651).

## Author Contributions

**Conceptualization:** Sara Brin Rosenthal, Hao Wang, Chi-Hua Chen.

**Data curation:** Cin Liu.

**Formal analysis:** Sara Brin Rosenthal, Hao Wang.

**Funding acquisition:** Chi-Hua Chen.

**Investigation:** Sara Brin Rosenthal, Chi-Hua Chen.

**Methodology:** Sara Brin Rosenthal, Chi-Hua Chen.

**Project administration:** Hao Wang, Chi-Hua Chen.

**Resources:** Chi-Hua Chen.

**Supervision:** Chi-Hua Chen.

**Validation:** Sara Brin Rosenthal, Hao Wang, Chi-Hua Chen.

**Visualization:** Sara Brin Rosenthal, Hao Wang, Chi-Hua Chen.

**Writing – original draft:** Sara Brin Rosenthal, Hao Wang, Linda K. McEvoy, Chi-Hua Chen.

**Writing – review & editing:** Sara Brin Rosenthal, Hao Wang, Da Shi, Cin Liu, Ruben Abagyan, Linda K. McEvoy, Chi-Hua Chen.

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
