## [Decision Letter · Decision Letter 0]

1 Oct 2021

Dear Dr. Wang,

Thank you very much for submitting your manuscript "Mapping the gene network landscape of Alzheimer’s disease through integrating genomics and transcriptomics" for consideration at PLOS Computational Biology.

As with all papers reviewed by the journal, your manuscript was reviewed by members of the editorial board and by several independent reviewers. In light of the reviews (below this email), we would like to invite the resubmission of a significantly-revised version that takes into account the reviewers' comments. In your revisions please in particular address the comments from Reviewer 1.

We cannot make any decision about publication until we have seen the revised manuscript and your response to the reviewers' comments. Your revised manuscript is also likely to be sent to reviewers for further evaluation.

Sincerely,

Feixiong Cheng, Ph.D.

Guest Editor

PLOS Computational Biology

Ilya Ioshikhes

Deputy Editor

PLOS Computational Biology

Reviewer's Responses to Questions

**Comments to the Authors:**

Reviewer #1: The manuscript titled ‘Mapping the gene network landscape of Alzheimer’s disease through integrating genomics and transcriptomic’ (Manuscript No: PCOMPBIOL-D-21-01541) integrated multi-omics data to explore the key landscape of AD, which offer new insight for understanding the pathophysiology of Alzheimer’s disease. Especially, the starting point is based on the AD risk genes from GWAS analysis on a larger number of AD-by-proxy samples and autopsy-documented AD samples.

I have some points to discuss with authors:

Major comments:

1. The pathways analysis is based on the expanded network after network propagation algorithm by AD risk genes from GWAS analysis. As mentioned in introduction, genetic variations captured by GWAS are inherited, whereas transcriptomic provides information on factors that are both inherited and non-inherited.

a. In table 1, it is interesting that the key pathways from expanded network module are not significant enriched when performing AD cortex RNAseq enrichment (BH FDR). The difference between genes from GWAS data and from transcriptomic data should be discussed. Or, further explanation could be given for that.

b. This work conducted network propagation algorithm based on AD risk genes from GWAS analysis and validate the obtained module clusters by significantly changed genes from transcriptomic data. What if network propagation algorithm was firstly performed on genes from transcriptomic data and then integrated with GWAS data? Will the final pathways and conclusions be different?

2. In section AD disease module of result, the manuscript declared that ‘Out of the 192 genes, 142 were found in the interactome (STRING database) and included in our subsequent analyses’. It means that 50 risk genes were excluded due to their missing in PPI network. However, it there possibility that some of 50 genes are the key genes to the pathophysiology of Alzheimer’s disease. If so, it might be one limitation of the work.

The authors had better provide which version of STRING database they utilized (The latest one or ?). In addition, the influence of some genes that cannot be mapped to interactome network should be discussed.

Minor comments:

1. In the transcriptomic study in the section methods, to compare the overlapped situation between significant regulated genes from transcript data and the genes in expanded AD disease module, hypergeometric test was used. The total number samples considered in the test should be given. Is it the number of all the genes in GWAS analysis or all the genes in interactome network?

2. In figure 5, mean expression (FPKM) was used to show the cell-specific gene expression in each cluster. With gene expression data, GSEA (Gene set enrichment analysis) could also be considered to calculated the enrichment score of each cell type in specific clusters.

Reviewer #2: This paper integrated AD GWAS data with protein-protein interaction network and identified 142 AD risk genes plus 646 network-proximal genes. They also used independent data such as mouse knockout to validate the functions of those genes. Further, they related their PPI-based AD gene network to the population gene expression data (temporal cortex, AD vs. health), and found 17 gene clusters enriched with various functions/pathways. Using those enriched pathways, they also found a hierarchical structure in their AD network, implying higher order sequential interactions among AD gene functions. Finally, they associated the clusters with recent single cell data and found microglial functions in the clusters such as immunoregulation. In general, the paper reads logically and is also well organized. However, I have the following concerns and suggestions:

● Under summary of molecular pathways of AD, the authors mention that the 5 classes of functionally related molecular pathways that are associated with AD could be changed slightly due to some pathways having multiple functions and being involved in various diseases. It may be helpful to indicate additional functions and how those 5 classes could change. Perhaps, providing this in supplementary information or elaborating a bit more could provide more informative context for the reader.

● In Introduction, the authors do mention 4 papers (25 to 28) which potentially explain evidence of pathological progression and different regional vulnerabilities in AD, but it may be helpful to elaborate a bit more on whether those studies utilized the cerebellum as a hypothesized control and temporal cortex as major disease-relevant tissue. Perhaps adding some more literature and context on the brain region-specific progression of AD would help motivate this approach. Maybe, for example, the cerebellum is never impacted in AD (and a line on its functioning in the brain and statistics on why it is never involved), can be powerful to include. For instance, while it is great that the authors have done literature review for brain regions in AD, it may help to also synthesize and weave in that literature (as it may be too tedious for the reader to go through the literature to validate the support).

● In Introduction, it may help to add a parentheses definition of multiscale community detection to help the reader better understand.

● While the authors mention a lot of citations to support methodology (that is great and does boost confidence in the approaches), it may be helpful to elaborate on what “poorly characterized or new genes are”, especially since the authors removed such genes (please see AD disease module section, end of 1st paragraph, under results).

● From the manuscript, it is implied that 50 AD risk genes are excluded (that were previously identified in the large AD-by-proxy GWAS). Is there any other analysis that was done using these genes? Could there be other interactome data including these genes that may have been considered. What are the risks involved or cons of this approach? The authors only mention a very brief limitation in the discussion (i.e., “There are multiple choices of interactome, each with their own pros and cons, and many tissue-specific networks that can be obtained”). They could potentially add in details on losing the 50 AD risk genes using STRING interactome as a potential limitation. Could the genes have been uncovered in other interactomes? Elaborating more about the potential limitations can help as well and explaining more about the “pros and cons” of tissue-specific networks can help. In addition, authors may consider adding a few more details on why the STRING interactome was selected (i.e. what were the results of a systemic evaluation of the performance of diverse networks).

● It is a bit confusing why the authors first use STRING interactome for the analysis (if only 142 AD GWAS risk genes are identified instead of the 191 that are identified in GIANT). Why replicate the analysis using GIANT to test consistency of the localization analysis and not the other way around, for example? Are the results of STRING in other applications much better than those of GIANT?

● Under “Network propagation”, the authors mention that this process “simulates how heat would travel, through a network starting from a set of seed nodes.” What does heat represent (in terms of exploring the network proximity to a set of genes identified as significantly associated with AD), and why is this important?

● In Discussion, the authors mention that the “dataset for AD risk genes from GWAS is incomplete”, but what does this mean and how can it be more complete?

● In Fig. 5B, the authors mention that the inner circles are classes of identified pathways that are functionally related. What does the size of the inner circles exactly mean? It may be helpful to add a legend or a little more explanation to the figure caption. Also, if “Immune reactions” is right next to Microglia (which is expected as Microglia are immune cells in the brain) does this mean that it is only in Microglia (and same for Substance metabolism)? If that is the case, then about Endothelial cells as they don’t have 1 unique circle in front of them. What determines the size of each cell type in the outer circle? In other words, this “circos-like” plot can be a bit confusing and may need to be re-thought as it may currently be difficult for the reader to appreciate the novel findings. The main text only mentions 1 line for the figure “A schematic model of gene network in AD pathophysiology based on results of the present study is illustrated in Fig. 5B”, which does not explain much either.

● In terms of visuals, Fig. 2 was superb in terms of detail, annotation, and legends, and innovation. Nonetheless, in the caption, the authors could add some example of proposed drug targets that have the bold black outlines. They could consider saying, (e.g. CHRNA2 and PTK2B are bolded examples). Also, what does the font size mean: why does CHRNA2 have a larger font size than TPM1 or ACTA2, for example? This explanation of font size can be added to Figure 2.

● Some of the other figures can be similar to Fig. 2 in terms of the approach. For instance, in Fig. 3, it may help to provide a legend (similar to the “cluster size” annotation in Fig. 2) to explain how the circle height relates to the # of genes that result in the phenotype when knocked out in mice. That is, for example, is the small circle just 1 gene and the large circle 100 genes? Fig. 4 caption explains pie chart colors but still having a legend (similar to Fig. 2 “DE Beta Statistic”) explaining the colors can help.

● Some of the methods may need more explanation, especially in terms of the math. What is alpha in “Network propagation”. Where do we get the heat vector at time T (for F^t) and the value of Y? It may help if there is supporting code that is publicly-available for the readers to go through (i.e. an R markdown file or ipython notebook tutorial) to help them better understand how the network methods were exactly implemented

● The authors also mention trends that are not statistically significant under the section “Cell-specific preferential expression”. It may help to report p-values to better understand if the results were close to statistical significance. Also, providing more biological interpretation of the cell-specific results would be helpful as the authors provide references to Figures and Supplements but could also list them out in the results. For instance, microglia have been recently implicated in AD neuroinflammatory processes.

● Under the statement: “We observed a cluster of genes associated with the ZAG-PIP complex”, it may be helpful to list out these genes that are involved in the ZAG-PIP complex. It appears as though ZAG-PIP complex is #15 module in Fig. 2 but there are no distinct clusters of genes there that point to it. The same for RNA metabolic process, which has no module of genes pointing to it but has some mention of genes like ZC3HAV1, DNAJC2, SYMPK, etc. in the Discussion section.

● It was great that the authors mention the different pathways in the Discussion section and the biology of the findings. It may be helpful to cite the Figures and supplementary figures associated with the different pathways. For instance, it is unclear where we could find the “cluster of 33 genes” identified by the data-driven approach.

● The authors mention Fig 4A under Proteostasis, but there is no figure 4A.

● The organization of the discussion is unclear, nonetheless. The authors mention potentially clinically-relevant clusters to discuss, and it appears as though those could belong to Fig. 2, but the authors do not really follow a flow for Fig 2. For instance, where is proteostasis in Fig 2? It is a section in the discussion but not really mentioned in the figure. Also, Complement Activation is mentioned in Discussion, and is #3 cluster (Red) in Fig. 2. Please clarify the logic for the Discussion. It may be helpful to allude to the pathways based on Fig 2 (or respective figure) along with the cluster #. Another example is APOE and Lipid Metabolism, which seems to potentially come from Cluster # 7 (purple, lipoprotein regulation) in Fig. 2; in that case, it may help to use consistent subheading with the cluster name or annotate/explain the location of the corresponding figure.

● It may help to really emphasize key findings in a paragraph in “Summary of Molecular Pathways in AD” as there is a lot going on in this paper (which is all great work), but the reader may be left wondering what to fully take away from this study. For instance, is this an approach that can be used in other diseases/contexts? Is this shedding light on future research in this area and where do we go from here? What can biologists grasp from these modules and findings? What is the key importance of these 5 classes of functionally-related molecular pathways that are significantly associated with AD? These findings are useful but it felt like they were not presented to the reader with enough justification of their importance. What is the future work?

Minors

● Grammatical errors and typos:

○ For example, under “Summary of Molecular Pathways in AD”, the first sentence contains an error, where “of” should probably be removed.

○ Another is in the introduction for the sentence: “We the integrate data of transcriptomic dysregulation in AD brains with the expanded AD disease module…”. Here, “the” should be replaced with “then”

○ In Fig. 5B, “Endothelia” could be a typo for “Endothelial”. If not enough space, perhaps abbreviating the name can help.

○ In Fig 3. Perhaps the x-axis label could be “log(OR) +/- 95% CI” with the / added between + and -.

○ Under Summary of Molecular Pathways in AD, 1st sentence may need hyphen “-” between functionally related, so it is: functionally-related

● Fig. 1 caption only mentions 2 words (Analysis workflow) and it may help to paraphrase the figure and the connections again so that the reader can recall the workflow. It may make the figure friendlier for readers then, as some could rely on the caption to help summarize the figure.

● In the introduction, it is mentioned that the latest large AD GWAS have identified over 40 risk loci. It is unclear if this is 1 GWAS or multiple GWAS (i.e. Jansen et. al, Kunkle et. al, etc.) SNPs combined. There was also a recent AD GWAS that has been published. The authors had 7 different references to GWAS studies for AD for this sentence, making it a bit confusing. It may help to refer to AD GWAS as “Jansen et. al”, “Kunkle et. al”, for instance. Again, a minor point, but it may be informative for readers.

● Figure 3: it is really great and a reader-friendly figure with few key terms, which is really thoughtful. Nonetheless, it seems as though abnormal synaptic transmission and abnormal CNS synaptic transmission are very similar (even visually, in terms of log(Odds Ratio) +/- 95% CI) and from the terms, perhaps 1 of the terms is a subset of the other. It may be better to include another term instead or remove altogether or provide more context on why both were included.

● This may be minor, but the number selected for random samplings could be consistent across approaches or explained more. For instance, why were 5,000 gene sets randomly selected to build up the null distribution for Network Propagation but only 1,000 random samplings used to build up the distribution for network localization? Is it due to computational resource constraints?

● Under network propagation, is z > 2 meant to be z > 1.96 (significance level of 5%) given that these are network propagation z-scores? Otherwise, what is the intuition for the cut-off of z > 2 for chance?

● In the Fig. S2 heatmap, the clustering and labeling is great, but it is a bit difficult to read the gene names properly. More height for the heatmap would help. A great resource in R could be the complexHeatmap package, which can really make beautiful heatmaps with annotations and clustering as well. Moreover, complexHeatmap could make the division in the heatmap between healthy and controls more prominent and easier to visualize.

● Sharing the code on github or a publicly-available location can greatly help.

● Instead of listing the URL for Agora directly (in section: “Transcriptomic dysregulation proximal to AD genes”), it may help to just cite it as the authors provide the full URL elsewhere in the methods.

● May be helpful to provide a citation for “Involvement of the cholinergic system in AD has been known since the 1970s”.

● May also consider adding in GWAS p-values if it helps motivate findings.

Reviewer #3: The paper utilized a network propagation methods to build AD disease module with GWAS inputs. After that combing with transcriptome data, the authors proved that the identified AD disease module are enriched with multiple disease-associated pathways. And cell-specific expression data demonstrated that genes in AD disease module are enriched with microglia expressed immune and complement activation functions related genes.

I think the overall manuscript design is rigorous and systematic.

1. At the end of page 21 and beginning of page 22, for the network propagation formula, there is a parameter alpha in equation, can the author provides some metric to explain which alpha value you use and why you use this alpha value?

2. Second paragraph of page 22: node z score computation, in the method description, the authors mentioned node z score, however do NOT mention the corresponding p value, has the author considered p value as well or not (just z score)?

3. The final formed AD disease module includes 788 genes with 142 risk genes and 646 network-proximity genes (non-GWAS evidenced). The authors have systematically discussed the 142 risk genes, however do not talk too much about the rest 646 network proximity genes. The authors could provide some statistical evidences for these 646 network-proximity genes by utilizing some public available AD-knowledge database, e.g., the open target platform, DisGeNET, to demonstrate that compared to randomly selected 646 genes, the 646 genes you predicted are more AD-associated. in this way, it could make people better believe that your method does have the capability to predict potential or likely AD-associated risk genes from GWAS inputs or the network module you found could provide useful information. I just think this could also help improve your manuscript novelties.

**Have the authors made all data and (if applicable) computational code underlying the findings in their manuscript fully available?**

Reviewer #1: Yes

Reviewer #2: **No: **open-source codes are missing for reproducibility

Reviewer #3: **No: **no code provided

PLOS authors have the option to publish the peer review history of their article (what does this mean?). If published, this will include your full peer review and any attached files.

Reviewer #1: No

Reviewer #2: No

Reviewer #3: No
---

## [Decision Letter · Decision Letter 1]

25 Jan 2022

Dear Dr. Wang,

Thank you very much for submitting your manuscript "Mapping the gene network landscape of Alzheimer’s disease through integrating genomics and transcriptomics" for consideration at PLOS Computational Biology. As with all papers reviewed by the journal, your manuscript was reviewed by members of the editorial board and by several independent reviewers. The reviewers appreciated the attention to an important topic. Based on the reviews, we are likely to accept this manuscript for publication, providing that you modify the manuscript according to the review recommendations.

Please take careful the minor comments from the Reviewer 2# and try to submit the revised manuscript in 2-4 weeks.

Sincerely,

Feixiong Cheng, Ph.D.

Guest Editor

PLOS Computational Biology

Ilya Ioshikhes

Deputy Editor

PLOS Computational Biology

[LINK]

Please take careful the minor comments from the Reviewer 2# and try to submit the revised manuscript in 2-4 weeks.

Reviewer's Responses to Questions

**Comments to the Authors:**

Reviewer #1: The responses from authors have answered my questions well.

Reviewer #2: The authors have addressed most of my concerns. I just have some suggestions as follows.

6. The remark by the authors to point 1b is helpful and it is duly noted that presenting an alternative network could complicate the paper. Perhaps the limitations section could incorporate this limiting factor and mention in future work other networks could be considered. The authors could still reiterate the power of the STRING interactome as being the most robust interactome while still presenting the limitations of the approach.

7. It is great that the authors added in a paragraph describing the network propagation and what. It is helpful to perhaps elaborate a little further on what it means for heat to diffuse through a network, as that may be unclear still; at the moment, it sounds like a physics scenario with terms like heat, diffusion, etc. Nonetheless, it is great that the authors provided this helpful paragraph.

9. Revision made to the figure caption. Nonetheless, it may help to add these same comments and clarifications (e.g. what was mentioned about Microglia and “immune reactions” and “substance metabolism”) to the figure text or the manuscript (or some supplementary documentation) as this clarification could help potential readers better understand the plot.

Reviewer #3: I checked my review feedback modifications; I am satisfied with all answers. Great work, I also learned multiple things.

**Have the authors made all data and (if applicable) computational code underlying the findings in their manuscript fully available?**

Reviewer #1: Yes

Reviewer #2: None

Reviewer #3: Yes

PLOS authors have the option to publish the peer review history of their article (what does this mean?). If published, this will include your full peer review and any attached files.

Reviewer #1: No

Reviewer #2: No

Reviewer #3: No

Figure Files:

Data Requirements:

Reproducibility:

References:

---

## [Editor Report · Decision Letter 2]

8 Feb 2022

Dear Dr. Cheng,

We are pleased to inform you that your manuscript 'Mapping the gene network landscape of Alzheimer’s disease through integrating genomics and transcriptomics' has been provisionally accepted for publication in PLOS Computational Biology.

Before your manuscript can be formally accepted you will need to complete some formatting changes, including improvement of figure's resolution, which you will receive in a follow up email. A member of our team will be in touch with a set of requests.

Best regards,

Feixiong Cheng, Ph.D.

Guest Editor

PLOS Computational Biology

Ilya Ioshikhes

Deputy Editor

PLOS Computational Biology

Comments from the Editors: The authors are highly suggested to improve the quality and resolution of all main figures during the production stages.

---

## [Editor Report · Acceptance letter]

18 Feb 2022

PCOMPBIOL-D-21-01541R2 

Mapping the gene network landscape of Alzheimer’s disease through integrating genomics and transcriptomics

Dear Dr Wang,

I am pleased to inform you that your manuscript has been formally accepted for publication in PLOS Computational Biology. Your manuscript is now with our production department and you will be notified of the publication date in due course.

With kind regards,

Anita Estes
